# Optimal Design of Seismic Resistant RC Columns

**DOI:** 10.3390/ma13081919

**Published:** 2020-04-19

**Authors:** Paolo Foraboschi

**Affiliations:** Dipartimento Culture del Progetto, Università IUAV di Venezia, Dorsoduro 2206, 30123 Venice, Italy; paofor@iuav.it

**Keywords:** capacity design, minimum cross-section, strength hierarchy, strong column–weak beam, thin RC column

## Abstract

Although the author is well aware that it is nothing special, presented here is the method that he uses to design the columns of a seismic resistant reinforced concrete structure, in hopes that this could be of use to someone. The method, which is directed at satisfying the capacity design requirements without excessively large sections, consists of proportioning the column so that the seismic action effects shall be resisted by the maximum of the bending moment–axial force interaction curve. That design condition is defined by two equations whose solution provides the optimal aspect ratio (or, alternatively, the optimal section side length) and the maximum feasible reinforcement ratio. The method can be used directly to determine the optimal column for given beam spans and vertical loads, or indirectly to determine the optimal beam spans and vertical loads for given cross-sectional dimensions. The paper presents the method, including its proof, and some applications together with the analysis on the optimality of the obtained solutions. The method is intended especially for the practicing structural engineer, though it may also be useful for educators, students, and building officials.

## 1. Introduction: Framing the Subject Matter and Review of Literature

When reinforced concrete (RC) frames are used as part of a seismic-resisting system in buildings that are designed to resist earthquakes, special requirements must be satisfied. Beams, columns, and beam–column joints in RC frames must be proportioned and detailed to undergo extensive inelastic deformations as well as to resist flexural, axial, and shearing actions that result as a building sways through multiple displacement cycles during strong earthquake ground shaking.

Special proportioning and detailing requirements result in a frame capable of resisting extreme earthquake shaking without global collapse, and of resisting the design seismic action for local failure prevention (life safety), retaining its vertical load-carrying capacity as well as a residual seismic load-carrying capacity after the seismic event. Those frames can be called “special frames” or “intermediate frames” (the difference is explained below), because of those additional requirements, which improve the seismic resistance in comparison with less stringently detailed frames, called “ordinary frames”.

The capacity of special frames to resist large seismic events is produced by strength and dissipation (dissipative behavior). Accordingly, the design of special frames shall provide the structure not only with substantial horizontal load-carrying capacity but also with substantial capacity to dissipate energy without significant reduction of its overall resistance against horizontal and vertical loadings. Thus, design requirements are prescribed by seismic codes for special moment frames. The proportioning and detailing requirements for special moment frames are intended to ensure that the inelastic response is ductile (capacity design).

The capacity of intermediate frames to resist large seismic events is only produced by strength (non-dissipative behavior, essentially elastic). Accordingly, the design of intermediate frames shall provide the structure with adequate horizontal load-carrying capacity (which, all the rest being equal, is substantially greater than if the behavior was dissipative). Thus, only strength requirements are prescribed by seismic codes for intermediate moment frames.

Ordinary frames are not only the frames in buildings that are not designed to resist earthquakes (located in zones with no seismic activity), but also the frames in buildings designed to resist earthquakes but not used as parts of the seismic-resisting structural system of the building (secondary seismic members or elements; gravity-only frames). While the former ones do not have to satisfy any seismic provisions, the latter ones shall be designed and detailed to maintain support of gravity loading when subjected to the displacements caused by the most unfavorable seismic design condition.

Intermediate frames are easier to design and verify than special frames, but often entail columns with larger cross-sectional dimensions, especially if the design seismic action is not low. In fact, providing the column with the required horizontal force strength is often more demanding than providing details that enable ductile flexural response in yielding regions. 

This paper focuses on special RC frames, but the method that is presented can also be applied to intermediate frames.

### 1.1. Capacity Design and Strength Hierarchy: Strong Column–Weak Beam Rule

The majority of the seismic design rules for special frames are rooted in the concept of “capacity design”, which is a design approach composed of basic principles that constrain the structure to perform in certain desirable manners. The principles of capacity design require both detailing ductile components so that they can sustain the large deformations that may occur under strong earthquakes without a significant loss of strength (hysteretic energy dissipation) and providing the non-ductile components with sufficient strength to remain essentially elastic throughout the large deformations that may occur (strength hierarchy) [1,2,3,4,5,6].

Hysteric dissipation of energy converts the kinetic energy of motion into heat. Accordingly, each member selected to deform inelastically must be suitably designed and detailed, so as to guarantee the energy dissipation in the member by means of stable, wide, and compact hysteretic loops, over the range of displacement amplitudes expected in the seismic design situation [7,8,9].

Hierarchy of strength of the various structural components and failure modes ensures a suitable plastic mechanism and avoids brittle failure modes. Accordingly, each member selected to deform elastically is given an appropriate strength to ensure that, when required, only the chosen plastic mechanisms can develop within the structure, thus ensuring desirable and predictable inelastic behavior during an extreme seismic event [10,11,12,13,14].

Strength hierarchy is achieved when the members that are not part of the energy dissipating system have strength equal to or greater than the overstrength capacity of the primary energy dissipating members. In structural design practice, the strength hierarchy is guaranteed using an overstrength factor to define the maximum capacity that a primary energy dissipating member may reach beyond the nominal design capacity (overstrength capacity) [15,16,17].

The hierarchy of strength includes some rules. One of the main rules is the strong column–weak beam provision, called “minimum flexural strength of columns” in ACI 318 seismic provisions [18] and “column-beam moment ratio” in AISC seismic provisions [19], while Eurocode 8 does not make use of any specific collocation [20]. That provision specifically refers to frames, checking that lateral loads will cause yielding in beams rather than in columns. It is intended to avoid hinge formation in multiple columns at the same level, which could cause story collapse, as well as to ensure an overall dissipative and ductile behavior that spreads inelastic response over several stories [21,22,23,24].

The strong column–weak beam provision shall be guaranteed by deriving the design action of the columns at a joint from equilibrium conditions, assuming that plastic hinges with their possible overstrengths have been formed in the beams at the joint. Namely, as previously mentioned, the plastic moment of a beam is taken as its design plastic moment multiplied by an overstrength factor (overstrength plastic moment). Accordingly, the columns at a node shall be designed to resist the bending moments generated by the overstrength plastic moments of the beams at the joint [2,12,13,14].

Another essential rule in order to obtain the strength hierarchy is to avoid shear failure, because shear failure, especially in columns, is relatively brittle and can lead to rapid loss of lateral strength and axial load-carrying capacity. However, this rule does not usually dictate the cross-sectional dimensions of the columns, so it is herein ignored [1,5,7,9,11,12,13,25,26].

### 1.2. Columns of Special Frames

The frames free internal subdivision of their load-bearing function, so it is the means of allowing freedom in the organization of space. For that reason, the frame is often preferred over other structural types. The frame is also selected as the structural type when architectural space planning flexibility is desired. Those architectural requirements push to select the frame as the seismic-resisting system as well, although other structural types perform better under seismic loading (e.g., shear walls).

Those architectural requirements also push to get the columns as small as possible. Unfortunately, that desire clashes with the strong column–weak beam provision. For RC frames [1,3,16,18,25,26,27], actually, the strong column–weak beam provision is very demanding from an architectural perspective, since its satisfaction requires columns with large cross-sections [13,15,28,29,30,31,32,33,34,35].

As a matter of fact, the design of the beams is usually dictated by the ultimate limit state design situation (static load-carrying capacity). Thus, the design of the columns is dictated by overstrength plastic moments that are dictated by the static (vertical) loads and not by the seismic (horizontal) loads. That logical sequence, which is intrinsic to strength hierarchy, implies that the cross-sectional dimensions of the columns do not depend on the design seismic action. Accordingly, if the beam spans or/and floor dead and live loads are not small or moderate, the structural design implies large cross-sectional dimensions of the columns whatever the local seismic hazard.

While the selection of the frame may be at the discretion of the architectural designer, the design of the column is essentially at the discretion of the structural designer. Ergo, the structural designer must solve a problem, namely to design seismic resistant RC columns with the optimal cross-sectional dimensions. Unfortunately, no method is available in the literature to optimize the column.

That problem can be split into two sub-problems, which are (1) to optimize each beam (2) and to optimize the column given the beams. Hereinafter, it is assumed that the structural designer has taken care to minimize excess capacity of the beams or that the beams have been designed according to other criteria (e.g., the relationship with the floor). The paper is devoted to solving the second problem: it presents a method to design RC columns that satisfy the strong column–weak beam provision with the minimum impact on the structural and architectural design.

## 2. Bending Moment–Axial Force Interaction Curve as a Criterion for Optimality

The method refers to the simultaneous bending and axial capacity of a section, namely to the ultimate combinations of bending moment and axial force [1,2,4,5,6,8,10,13,28,29,32,35,36,37,38,39,40,41,42,43]. Those combinations define the bending moment–axial force interaction curve. Each ultimate combination is denoted by *M_u_* and *N_u_*.

The bending moment–axial force interaction curve has an outward curvature, which implies an increasing trend up to a peak and a decreasing trend up to zero. Accordingly, the bending moment starts from the pure bending capacity (the point whose abscissa is zero), reaches a maximum value for a certain value of the axial force, and is equal to zero for the pure axial force capacity (the point whose abscissa is the maximum axial force of the domain).

The method that is presented herein employs the maximum bending moment of that domain. Therefore, the first part of this paper is devoted to defining the value of that maximum, that is to say, the abscissa and ordinate of the peak of the interaction curve.

The directions of the vectors of forces and couples, as well as whether stresses and strains are compression or tension, are known in advance (Figure 1 and Figure 2). Thus, each mathematical expression only accounts for the magnitude of a quantity, while it disregards its sign, which is that exhibited by the figures. In other words, the mathematical expressions deal with the absolute value of each quantity.

### 2.1. Reference System and Mechanical Assumptions

The reference system (Figure 1) is a rectangular RC cross-section whose height and width are denoted by *H* and *B*, respectively. That cross-section is cut from a member under combined bending moment and axial force, where both these actions are substantial. Hence, the reference system may be the cross-section of an RC column, which is part of a frame that is subjected to a lateral load due to a seismic action. That is to say, what follows holds true for any RC element, but herein is only used for the column.

As shown in Figure 1, the area of the tension longitudinal steel reinforcement is denoted by *A_s_*, and the area of the compression longitudinal steel reinforcement is denoted by *A^’^_s_*. When the outcome of these developments is applied to a column, *A^’^_s_* = *A_s_* as the behavior is symmetric. For the sake of generality, however, that condition is not considered in this section of the paper.

The gross concrete cover is denoted by χ (i.e., χ is the distance from extreme tension/compression fiber to centroid of tension/compression reinforcement). When the outcome of these developments is applied to a column, the above-mentioned symmetry implies that there is no difference between the concrete cover of the tension and compression reinforcements. For the sake of simplicity, however, the difference in concrete cover is not considered in this section of the paper, since it is always slight.

The effective depth of the RC cross-section is denoted by *d* (i.e., *d* = *H* − χ), and the depth of the neutral axis by *y*. Concrete crushing strain is denoted by ε*_cu_*, concrete compression strength by *f_c_*, steel yielding strain by ε*_y_*, steel yielding stress by *f_s_*, and steel modulus of elasticity by *E_s_*. Concrete maximum compression strain, compression reinforcement strain, and tension reinforcement strain are denoted by ε*_c_*, ε^’^*s*, and ε*_s_*, respectively (Figure 1).

As shown in Figure 2, the resultant compression force in the concrete is denoted by *C*, and the distance of *C* from the centroid of the section is denoted by λ*_c_*. The resultant compression force in the steel is denoted by *F^’^_s_*, and the distance of *F^’^_s_* from the centroid of the section is denoted by λ^’^*_s_*. The resultant tension force in the steel is denoted by *F_s_*. The distance of *F_s_* from the centroid of the section is denoted by λ*_s_*.

Failure of the RC section can be caused by either the concrete crushing, the reinforcing bars debonding, or the reinforcing bars fracturing. Hence, the pivotal point in the analysis can occur at either ε*_cu_*, debonding strain, or ultimate steel strain, whichever occurs first.

Each bar is supposed to be anchored and adequately overlapped. Thus, the bar attains *f_s_* before debonding. It follows that reinforcing bar debonding cannot occur. The ultimate strain of steel is more than 30 times greater than ε*_y_* and more than 20 times greater than ε*_c_*. It follows that the concrete crushing can be assumed to occur before the reinforcing bar fracturing (i.e., the steel behaves either elastically or plastically, but does not fracture). Accordingly, the pivotal point of the ultimate strain profile is at ε*_cu_*.

The first assumption is that ε*_cu_* = 3.5‰. On one hand, this is a common assumption [18,20], but, on the other hand, it is a rough approximation. This assumption plays no role in what is proven below but plays a part in defining the optimal solution. However, this assumption can be removed, and the method can be used adopting different values of ε*_cu_*, in particular ε*_cu_* = 3.5‰, 2.7‰, and 4.5‰. In so doing, this assumption is overcome.

The second assumption is about the stress profile in the concrete. The resultant internal compression force in the concrete *C* is calculated using the equivalent rectangular concrete compressive stress block, which is a common assumption. The depth of the stress block is equal to 0.8⋅*y* (Figure 2).

The third and last assumption is that the cross-section remains plane after deformation, which is a common assumption as well.

### 2.2. Review of the Balanced Flexural Failure Condition

The failure condition for which the strain of the tension steel is equal to ε*_y_* is called balanced failure. In other words, the balanced failure is the ultimate condition for which the strain profile goes from ε*_cu_* at the compression edge to ε*_y_* at the tension steel. Since the mechanical assumptions used herein (Section 2.1) are those commonly used in RC analysis, the formulas that describe the balanced failure can be borrowed from classical literature.

The depth of the neutral axis *y* can be obtained using the third assumption.
(1)$$y\;=\;\frac{\varepsilon_{cu_{}}}{\varepsilon_{y}\;+\;\varepsilon_{cu}\;}\cdot d$$

The resultant internal compressive force *C* can be obtained using the second assumption.
(2)$$C\;=\;0.8\cdot B\cdot y\cdot f_{c}$$

The distance of *C* from the centroid of the section, which has been denoted by λ_*C*_, is
(3)$$\lambda_{C}\;=\;\frac{\;H\;}{2}\;-\;0.4\cdot y$$

The resultant internal compressive force in the steel, which has been denoted by *F*
^’^*_s_*, is
(4)$$F_{s}^{\prime }\;=\;\frac{\;y\cdot \varepsilon_{cu_{}}\;-\;\chi \cdot \varepsilon_{cu_{}}}{\;y\;}\cdot E_{s}$$
provided that *F*
^’^*_s_* from Equation (4) is lower than *A*^’^*_s_*⋅*f_s_*. Otherwise, *F*
^’^*_s_* = *A*^’^*_s_*⋅*f_s_*.

The distance of *F*
^’^*_s_* from the centroid of the section, which has been denoted by λ^’^_*s*_, is
(5)$$\lambda_{s}^{\prime }\;=\;\frac{\;H\;}{2}\;-\;\chi$$

By definition of the balanced failure, the resultant internal tension force in the steel, which has been denoted by *F_s_*, is
*F_s_* = *A_s_*⋅*f_s_*(6)

The distance of *F_s_* from the centroid of the section, which has been denoted by λ*_s_*, is
(7)$$\lambda_{s}\;=\;\frac{\;H\;}{2}\;-\;\chi$$

The axial force and bending moment at the balanced failure, which are, respectively, denoted by *M_ub_* and *N_ub_*, are obtained from the translational and rotational equilibrium of the section internal forces. Those equilibria yield:(8)$$N_{ub}\;=\;C\;+\;F_{s}^{\prime }\;-\;F_{s}$$
(9)$$M_{ub}\;=\;C\cdot \lambda_{C}\;+\;F_{s}^{\prime }\cdot \lambda_{s}^{\prime }\;+\;F_{s}\cdot \lambda_{s}$$
where *C*, λ_*C*_, *F*
^’^_*s*_, λ^’^*_s_*, *F_s_*, and λ*_s_* are provided by Equations (2)–(7), in which *y* is given by Equation (1).

### 2.3. Maximum Bending Capacity of a Reinforced Concrete Cross-Section

It can be proven that *M_ub_* from Equation (9) is the maximum internal bending moment that a given RC cross-section can make available. Accordingly, the balanced failure is the condition for which a specified RC cross-section reaches the peak of the bending moment–axial force interaction curve. The demonstration, which is basic, is given below.

A small (infinitesimal) counterclockwise rotation of the strain profile around the pivot (i.e., around ε*_cu_*) increases the strain in the tension steel and decreases *y*. The tension steel was at the plastic threshold, so the strain profile rotation makes the tension steel strain surpass ε*_y_*. However, the resultant internal tension force in the steel reinforcement remains *F_s_* given by Equation (6), because the tension steel was already in the plastic state. The decrease in *y* implies a decrease in the strain of the compression steel ε^’^*s*. It follows that the resultant internal compression force *F ^’^_s_* in the steel either decreases or remains the same, depending on whether ε^’^*s* ≤ ε*_y_* or ε^’^*s* > ε*_y_*. The decrease in *y* also implies a decrease in the resultant internal compressive force *C* in the concrete but also an increase in the distance λ*_C_* between *C* and the centroid of the section. However, the product *C*⋅λ*_C_* of Equation (9) decreases, since the lower *y*, the lower the area of the stress block (i.e., the stress block loses a portion of area).

Ultimately, a counterclockwise rotation of the strain profile around the pivot leads to a bending moment that is lower than *M_ub_*. Of course, that rotation also modifies *N_ub_*, which decreases.

A small (infinitesimal) clockwise rotation of the strain profile around the pivot (i.e., around ε*_cu_*) decreases the strain in the tension steel and increases *y*. The tension steel was at the plastic threshold, so the strain profile rotation makes the tension steel strain be lower than ε*_y_*. As a result, the tension steel undergoes a phase change from the plastic state to the elastic state, which implies that the resultant internal tension force in the steel decreases with respect to the value given by Equation (6). The increase in *y* implies an increase in the strain of the compression steel ε^’^*s*. It follows that the resultant internal compression force *F ^’^_s_* in the steel either increases or remains the same, depending on whether ε^’^*s* < ε*_y_* or ε^’^*s* ≥ ε*_y_*. In other words, *F ^’^_s_* does not decrease. However, the compression reinforcement is much nearer the pivot than the tension reinforcement, so the possible increase in *F ^’^_s_* is drastically lower than the decrease in *F_s_* (even for the column, whose compression reinforcement is equal to the tension reinforcement). The increase in *y* also implies an increase in the resultant internal compressive force *C* in the concrete but also a decrease in the distance λ*_C_* between *C* and the centroid of the section. The result of those variations is that the product *C*⋅λ*_C_* of Equation (9) increases, since the greater *y*, the greater the area of the stress block (i.e., the stress block gains a portion of area). However, apart from highly under-reinforced sections (which do not comply with codes), the decrease in the product *F_s_*⋅λ*_s_* is greater than the increase in the sum of the products (*F*
^’^*_s_*⋅λ^’^*_s_* + *C*⋅λ*_C_*).

Ultimately, a clockwise rotation of the strain profile around the pivot leads to a bending moment that is lower than *M_ub_*. Of course, that rotation also modifies *N_ub_*, which increases.

It is hence proven that the internal bending moment *M_ub_* and the internal axial force *N_ub_* at the balanced failure are the coordinates of the peak of the bending moment–axial force interaction curve. There are exceptions, which, however, do not belong to properly reinforced members.

### 2.4. Optimal Condition

Hereinafter, the balanced flexural failure is seen not only as the peak of the *M*–*N* ultimate interaction curve, but also as the condition that makes full use of the bending capacity of a given concrete cross-section and a given reinforcement. In fact, that condition provides the specified RC cross-section with the greatest strength against flexure.

A column that resists the design seismic action for the life safety requirement at the peak of the *M*–*N* ultimate interaction curve is herein defined as optimal. Accordingly, the criterion adopted to judge whether a column is optimal is whether the seismic action for the no-failure requirement is resisted by the maximum bending capacity that the cross-section can provide the column with, availing of the help of the axial force.

## 3. Expressing the Balanced Failure Equations in a Form Useful for Design

The method allows the design to choose either the aspect ratio of the section or the width of the column. That choice is typically made within the architectural design. If the aspect ratio of the column is established beforehand, then the width *B* can be expressed as a fraction α of *H*, i.e., *B* = α⋅*H* with 0 < α ≤ 1, where α is known. If the width is established beforehand, then α can be expressed as a function of *H*, i.e., α = *B*/*H*, where *B* is known. The presentation refers to α, which in either case is known.

As is common practice in structural design, the method allows the designer to choose the type of concrete and steel, as well as the concrete cover χ. It follows that the relevant mechanical parameters are known. As was previously mentioned, the structural behavior of an RC column is symmetric with respect to the bending axis, and so should be the reinforcement. Hence, in a column, *A^’^_s_* = *A_s_*. In the following mathematical expressions, which hold for the column (while the above expressions hold for any member), *A^’^_s_* is thus replaced by *A_s_*.
(10)$$y\;=\;\frac{\;\varepsilon_{cu_{}}}{\;\varepsilon_{y}\;+\;\varepsilon_{cu}\;}\cdot \left(H\;-\;\chi \right)$$

Equation (10) can be plugged into Equations (2)–(7) together with *B* and *d* expressed as a function of *H* (i.e., *B* = α⋅*H* and *d* = *H* − χ). As a result, Equations (2)–(7) are expressed as functions of *H* and *A_s_*. Plugging those formulas into Equations (8) and (9), the balanced flexural failure of the column section can be expressed as function of *H* and *A_s_*, while the other parameters belong to the data.

If the compression steel reinforcement has not surpassed the elastic threshold, Equations (8) and (9) eventually turn into
(11)$$\begin{array}{l} 0.8\cdot \left(\alpha \cdot H\right)\cdot \frac{\;\varepsilon_{cu_{}}}{\;\varepsilon_{y}\;+\;\varepsilon_{cu}\;}\cdot \left(H\;-\;\chi \right)\cdot f_{c}\;+\\ +\;A_{s}\cdot \frac{\left[\frac{\;\varepsilon_{cu_{}}^{}}{\;\varepsilon_{y}\;+\;\varepsilon_{cu}\;}\cdot \left(H\;-\;\chi \right)\;-\;\chi \right]\cdot \left(\varepsilon_{y}\;\overset{}{\underset{}{+}}\;\varepsilon_{cu}\right)\;}{\;{\left(H\;-\;\chi \right)}^{}\;}\cdot E_{s}\;-\;A_{s}\cdot f_{s}\;=\;N_{ub}\; \end{array}$$
(12)$$\begin{array}{l} 0.8\cdot \left(\alpha \cdot H\right)\cdot \frac{\;\varepsilon_{cu_{}}}{\;\varepsilon_{y}\;+\;\varepsilon_{cu}\;}\cdot \left(H\;-\;\chi \right)\cdot f_{c}\cdot \left[\frac{H}{2}\;-\;0.4\cdot \frac{\;\varepsilon_{cu_{}}}{\;\varepsilon_{y}\;+\;\varepsilon_{cu}\;}\cdot \left(H\;-\;\chi \right)\right]\;+\\ +\;A_{s}\cdot \frac{\left[\frac{\;\varepsilon_{cu_{}}^{}}{\;\varepsilon_{y}\;+\;\varepsilon_{cu}\;}\cdot \left(H\;-\;\chi \right)\;-\;\chi \right]\cdot \left(\varepsilon_{y}\;\overset{}{\underset{}{+}}\;\varepsilon_{cu}\right)\;}{\;{\left(H\;-\;\chi \right)}^{}\;}\cdot E_{s}\cdot \left(\frac{\;H\;}{2}\;-\;\chi \right)\;\\ +\;A_{s}\cdot f_{s}\cdot \left(\frac{\;H\;}{2}\;-\;\chi \right)\;=\;M_{ub} \end{array}$$

If the compression steel reinforcement has surpassed the elastic threshold and is in the plastic state, the balanced flexural failure of the column section is expressed by the following two equations, in lieu of Equations (11) and (12):(13)$$0.8\cdot \left(\alpha \cdot H\right)\cdot \frac{\;\varepsilon_{cu_{}}}{\;\varepsilon_{y}\;+\;\varepsilon_{cu}\;}\cdot \left(H\;-\;\chi \right)\cdot f_{c}\;=\;N_{ub}$$
(14)$$\begin{array}{l} 0.8\cdot \left(\alpha \cdot H\right)\cdot \frac{\;\varepsilon_{cu_{}}}{\;\varepsilon_{y}\;+\;\varepsilon_{cu}\;}\cdot \left(H\;-\;\chi \right)\cdot f_{c}\cdot \left[\frac{H}{2}\;-\;0.4\cdot \frac{\;\varepsilon_{cu_{}}}{\;\varepsilon_{y}\;+\;\varepsilon_{cu}\;}\cdot \left(H\;-\;\chi \right)\right]\;+\\ +\;2\cdot A_{s}\cdot f_{s}\cdot \left(\frac{H}{2}\;-\;\chi \right)\;=\;M_{ub} \end{array}$$

## 4. Design of the RC Column

The criterion according to which a seismic resistant RC column is optimal is that established in Section 2.4. That is, the concrete section and the steel reinforcement shall be dimensioned so that the design bending moment from the strong column–weak beam rule together with the design axial force from the analysis for the seismic design situation (life safety requirement) are the *M*–*N* coordinates of the peak of the bending moment–axial force interaction curve. In so doing, the column is proportioned so as to resist the seismic action with the greatest bending moment that is possible.

According to Section 2, the dimensioning of the column must be aimed at obtaining balanced failure. According to Section 3, the design of the column is dictated by either Equations (11) and (12) or Equations (13) and (14), depending on the state of the compression steel.

As already specified, the general design decides if the width is either a fraction of the height (aspect ratio) or has a given value (which is established). Whatever the case, the data of the problem include α. The structural design decides the type of concrete and steel. Thus, the data of the problem include ε*_cu_*, *f_c_*, ε*_y_*, *f_s_*, and *E_s_*.

The capacity design implies that the beams are designed before the columns. The design values of the bending moments and shear forces of the beams are obtained from the analysis of the structure for the seismic design situation (life safety requirement) and for the ultimate limit state design situation (static load-carrying capacity, which is frequently the most severe situation for a beam). Those values allow the beams to be designed. Consequently, the beams framing into the column are known. Thus, the data of the problem include the overstrength plastic moments of the beams at the joint with the columns.

The rotational equilibrium of the joint gives the end moment demand of a column, which is denoted by *M_R_* (Figure 3).

The strong column–weak beam rule is waived at the top level of multistory buildings. Thus, *M_R_* results from a rotational equilibrium with two resisting moments (that of the column under design and that of the column above) and two overstrength plastic moments (one, in the case of a corner or exterior joint).

The structural analysis carried out to design the beams also furnishes the column axial force in the seismic design situation for the considered sense of the seismic action, which is denoted by *N_E_*. Thus, the data of the problem include the axial force *N_E_* that the column has to resist together with the bending moment *M_R_* (Figure 3).

As a result, all the terms of Equations (11)–(14) are known, except the height of the concrete section *H* and the amount of tension steel reinforcement *A_s_*, which gives rise to a set of two simultaneous equations with two unknowns. The two equations are linear with respect to the two unknowns. Consequently, the system has one and only one solution, i.e., *H* and *A_s_*. The explicit form of the solution is not presented as it would have no use, because the solution can be obtained by any mathematical software.

Ultimately, Equations (11)–(14) give *H* and *A_s_*. Since the amount of tension steel reinforcement *A_s_* is equal to the amount of the compression steel reinforcement *A^’^_s_*, the solution defines the RC cross-section. That cross-section is the optimal design of the RC column.

## 5. Application of the Proposed Design Method

The method is here applied to two cases. Each case consists of designing a column of an RC plane frame. The top end of the column under design is connected to a joint into which two beams and an upper column frame. The column above is supposed to have the same cross-section and reinforcement as the column under consideration, as seen in Figure 3.

In both the case studies, the method is used in its direct way of treating the data and unknowns, namely, to determine the optimal column for given beams (spans and loads). In the first case, the method is also used in its indirect way, namely, to determine the optimal beam spans and vertical loads for a given cross-section.

Because the design of the column depends on the beams framing into the beam–column joint, the application examples refer to beams that have been sized to minimize excess capacity. Consequently, the results that are obtained in the following are optimal not only with respect to the given beams but also with respect the whole structural system.

The column longitudinal bars are located along the two sides parallel to the axis of flexure, i.e., the column longitudinal bars are not located around the perimeter of the column cross-section. The longitudinal bars are spaced far enough apart so that the concrete can easily flow between the bars. Minimum bar spacing is especially critical at splice locations. Splicing the vertical bars at every other floor eliminates some of the congestion but often creates construction difficulties. Thus, the application examples refer to vertical bars spliced at every floor and the splice location meets the minimum clear spacing between the longitudinal bars given in [18]. Longitudinal bar lap splices can be located either along the middle of the clear height or at the bottom of the column. Contrary to the former arrangement, the latter arrangement entails that the lap splices of longitudinal reinforcement are not positioned outside intended yielding regions but, on the other hand, it prevents placement errors. The author prefers the latter solution.

### 5.1. First Application Example

The geometry and steel reinforcement of the two beams that frame into the joint are shown in Figure 4. The design value of the concrete compressive strength (crushing stress) is *f_c_* = 16.7 N/mm^2^, and the design value of the steel strength (yielding stress) is *f_s_* = 391.3 N/mm^2^. The gross concrete cover is χ = 40 mm. The *B*/*H* ratio is α = 1. Hence, the column under design is square. The design axial force is *N_E_* = 1561.0 kN. The value of *N_E_* was obtained for a central column at the lowest level of a building made up of four stories plus the roof, with spans of 6.5 m.

The overstrength plastic moments of the cross-section of the beams (Figure 4) are *M_bl_* = 830.6 kN⋅m with the tension reinforcement at the top and the compression reinforcement at the bottom, and *M_br_* = 558.9 kN⋅m with the tension reinforcement at the bottom and the compression reinforcement at the top of the section. The demand of the column under design, which is given by the rotational equilibrium of the joint, is therefore *M_R_* = 694.8 kN⋅m.

Plugging those values into Equations (11)–(14), the system furnishes the solution. More specifically, by plugging in 1,561,000 for *N_ub_*, 694,800,000 for *M_ub_*, 1.0 for α, 3.5‰ for ε*_cu_*, 1.957‰ for ε*_y_*, 40 for χ, 16.7 for *f_c_*, and 391.3 for *f_s_*, the system defines *H* and *A_s_*.

The result is *H* = 450 mm and *A_s_* = 3716 mm^2^ (Figure 5). Summing up, the final design of the column consists of a square section of 450 mm sides, with steel reinforcement *A_s_* = *A^’^_s_* = 3716 mm^2^ at each side parallel to the axis of flexure. That amount of reinforcement can be obtained with seven bars with a diameter of 26 mm at each of the two edges. The seven bars can be placed on one row only, since the distance between two consecutive bars is sufficient for properly pouring the concrete and for achieving proper consolidation (Figure 5). The longitudinal reinforcement in particular does not result in congested splice locations.

The design must be completed with the reinforcing bars at the two sides perpendicular to the axis of flexure. Since the structural system is a plane frame, those bars are dictated by code limits for reinforcement of columns. If the length of those sides is less than 350 mm, it is not necessary to place any bars. Otherwise, it is enough to satisfy the minimum reinforcement prescribed by the code. In the case of a spatial frame, those bars depend on the relevant beams.

In order to show that the 450 by 450 mm column with 7 + 7 bars having diameter of 26 mm is the most favorable solution, two slightly varied concrete sections are analyzed.

The first alternative solution is the square section of side 400 mm, i.e., a slightly lower cross-section. If *H* = 400 mm, then *A_s_* must be no less than 5485 mm^2^. The best arrangement to obtain that amount of steel reinforcement is nine bars with diameters of 26 mm and one bar with a diameter of 30 mm, placed in two rows (so that the distance between the bars allows the concrete to be properly poured and to achieve proper consolidation). To recap, in order to simultaneously carry the axial force equal to 1561.0 kN and the bending moment equal to 694.8 kN⋅m, that section has to be reinforced with nine bars whose diameters are 26 mm and one bar whose diameter is 30 mm, at each of the two sides parallel to the axis of flexure.

Ultimately, if the side of the square cross-section is 50 mm lower, the amount of steel reinforcement that is necessary is about 1.5 times greater.

Considering also that those bars are both numerous and thick, that the concrete section must provide lodging for the lap splices, and that large diameter bars require long lap splices (more than 1.5 m), the 400 by 400 mm column is much less satisfactory than the 450 by 450 mm column.

The second alternative solution that is considered is the square section of side 500 mm, i.e., a slightly greater section. If *H* = 500 mm, then *A_s_* must be no less than 3001 mm^2^. The best arrangement to obtain that amount of steel reinforcement is seven bars with diameters of 24 mm, placed in one row. To recap, in order to simultaneously carry the axial force equal to 1561.0 kN and the bending moment equal to 694.8 kN⋅m, that section has to be reinforced with the same arrangement of reinforcement as the 450 by 450 mm column, while the diameter of the bars is 24 mm instead of 26 mm.

Ultimately, if the side of the square cross-section is 50 mm greater, the decrease in the amount of steel reinforcement is marginal.

Considering that the increase in size of the cross-section is substantial, the 500 by 500 mm column is much less satisfactory than the 450 by 450 mm column.

The proposed method also allows the general design to be calibrated in order to meet some intended goals. Especially, the method allows a desired cross-section to become the optimal by modifying the loads or/and the spans. In order to achieve that goal, the equations must be used differently; the height *H* is one of the data, together with the maximum reasonable amount of steel reinforcement that can be placed in this section, while the unknowns are *M_ub_* and *N_ub_*. 

The further step is to relate the end moment demand of the column *M_R_* and the axial force transmitted by the beams to the column *N_E_*, to the dead and live loads of the floor, and the spans of the beams. The last step is to modify (reduce) loads or/and spans so that *M_R_* is equal to *M_ub_* and *N_E_* to *N_ub_*.

Let us assume, for instance, that the architectural design requires a square column with side 400 mm. That section allows up to six bars with diameters of 24 mm to be placed at each edge, while a greater amount would be excessive. Thus, *A_s_* = 2714 mm^2^.

Plugging those values of *H* and *A_s_* into Equations (11)–(14), and solving the set of equations for *M_ub_* and *N_ub_* gives the bending moment and axial force at the balanced failure for which *H* = 400 mm and *A_s_* = 2714 mm^2^ are the optimal design solution. The result is *M_ub_* = 471.0 kN⋅m and *N_ub_* = 1249.3 kN. Those *M_ub_* and *N_ub_* can be obtained by decreasing either the total load by 19% or the spans by 7% (or else by modifying both of them, which allows each reduction to be lower). The requested reductions are not demanding and can be taken into consideration.

### 5.2. Second Application Example

The geometry and steel reinforcement of the two beams that frame into the node are shown in Figure 6. The design value of the concrete compressive strength (crushing stress) is *f_c_* = 20.0 N/mm^2^, and the design value of the steel strength (yielding stress) is *f_s_* = 391.3 N/mm^2^. The gross concrete cover is: χ = 40 mm.

In this second case, what is given is not the aspect ratio but the width, which is *B* = 300 mm. That value must be plugged into the equations in lieu of α⋅*H*.

The design axial force is *N_E_* = 1122.0 kN. The value of *N_E_* was obtained for a central column at the lowest level of a building made up of three stories plus the roof, with spans of 7.0 × 5.0 m.

The overstrength plastic moments of the cross-section of the beams (Figure 6) are *M_bl_* = 331.1 kN⋅m with the tension reinforcement at the top and the compression reinforcement at the bottom, and *M_br_* = 188.6 kN⋅m with the tension reinforcement at the bottom and the compression reinforcement at the top. The demand of the column under design, given by the rotational equilibrium of the joint, is *M_R_* = 260.0 kN⋅m.

Plugging those values into Equations (11)–(14), the system furnishes the solution. More specifically, by plugging in 1,122,000 for *N_ub_*, 260,000,000 for *M_ub_*, 300 for α⋅*H*, 3.5‰ for ε*_cu_*, 1.957‰ for ε*_y_*, 40 for χ, 20.0 for *f_c_*, and 391.3 for *f_s_*, the system can be solved for *H* and *A_s_*.

The result is *H* = 400 mm and *A_s_* = 1140 mm^2^ (Figure 7). Summing up, the final design of the column consists of a rectangular section of 300 × 400 mm sides with steel reinforcement *A_s_* = 1140 mm^2^ at each side parallel to the axis of flexure. That amount of reinforcement can be obtained with three bars with diameters of 22 mm at each of the two sides. The three bars can be placed on one row only, since the distance between two consecutive bars is sufficient for the pour and consolidation of the concrete in any zone of the column, including the zone along the lap-spliced bars (Figure 7).

The design must be completed with the reinforcing bars at the two sides perpendicular to the axis of flexure, which are dictated by code limits for reinforcement of columns.

In order to show that the 300 by 400 mm column with 3 + 3 bars having diameters of 22 mm is the most favorable solution, two slightly varied concrete sections are analyzed.

The first alternative solution is the 300 × 350 mm cross-section, i.e., a slightly lower section. If *H* = 350 mm, then *A_s_* must be no less than 1900 mm^2^. The best arrangement to obtain that amount of steel reinforcement is five bars with diameters of 22 mm, placed in one row. To recap, in order to simultaneously carry the axial force equal to 1122.0 kN and the bending moment equal to 260.0 kN⋅m, that section has to be reinforced by five bars with diameters of 22 mm at each of the two sides of the section parallel to the axis of flexure.

Ultimately, if one side of the rectangular cross-section is 50 mm lower, the amount of steel reinforcement that is necessary is about 1.7 times greater. 

Considering also that the space between those reinforcing bars is small and that the concrete section must provide lodging for the lap splices, the 300 by 350 mm column is less satisfactory than the 300 by 400 mm column.

The second alternative solution that is considered is the 300 × 450 cross-section, i.e., a slightly greater section. If *H* = 450 mm, then *A_s_* must be no less than 880 mm^2^. The best arrangement to obtain that amount of steel reinforcement is two bars with diameters of 20 mm and one bar with a diameter 18 mm, placed in one row. To recap, in order to simultaneously carry the axial force equal to 1122.0 kN and the bending moment equal to 260.0 kN⋅m, that section has to be reinforced with the same arrangement of reinforcement as the 300 by 400 mm column, but the diameters of the bars are 20 mm and 18 mm instead of 22 mm.

Ultimately, if the side of the square cross-section is 50 mm greater, the decrease in amount of steel reinforcement is marginal.

Considering that the increase in size of the cross-section is considerable, the 300 by 450 mm column is much less satisfactory than the 300 by 400 mm column.

## 6. Discussion

The application to real cases has proven that the method furnishes the optimal design solution in compliance with the capacity design principles, where optimal is meant in the sense that the column consists of the minimum cross-sectional size with the maximum reasonable amount of steel reinforcement. Optimality of the design solution may be for either given beam spans and vertical loads or given cross-sectional dimensions. The design solution, in the former case, consists of the cross-section and reinforcement, while in the latter case it consists of the beam spans and/or dead plus live loads. The application has also shown that the method is easily applicable and not time-consuming.

An amount of reinforcement greater than that of the optimal solution would engender a decrease in the section only if the increase in steel is large. However, such increase in steel reinforcement would imply a very small spacing between the bars, which does not allow the concrete to be properly poured (lap splices of the longitudinal reinforcement create a very congested area of the column as the number of vertical bars is doubled and the hoops must be tightly spaced). Such increase in steel reinforcement would also prevent the concrete to achieve good consolidation even if the contractor positioned internal vibration equipment prior to placing the reinforcement or used external vibration (if there was adequate access to all sides of the formwork). Moreover, such increase in steel reinforcement would imply that the bars cannot be properly placed in the field (not only the column cage but also the beam bars entering the joint, which must pass by each other and the column longitudinal bars).

A cross-section greater than that of the optimal solution would engender a significant decrease in the amount of reinforcement only if the increase in section size is substantial. However, such an increase in cross-section would imply a considerable impact on the architectural design.

Summing up, the possible variations lead to less satisfactory design solutions from the structural or architectural point of view, including the cost, since a column with a smaller cross-section creates steel congestion and a column with a larger cross-section saves a marginal amount of steel.

Architectural requirements often push to get the columns as small as possible, even less than the optimal design solution. The proposed method allows this architectural wish to be given consideration, as long as it is not too demanding (no more than 10%–15% of the side). The optimal solution can be slightly decreased provided that some actions are taken to counteract the negative effect of reinforcement congestion, namely splicing the vertical bars at every other floor, limiting maximum aggregate size, and either having a concrete mixture with a high slump or, even better, using self-consolidating concrete.

The axial force capacity of the section without bending moment is substantially greater that the design axial force from the analysis for the seismic design situation of the column (life safety) multiplied by the live load factor prescribed for the ultimate limit states (and thus, also multiplied by the code-prescribed dead load factor). The design bending moment from the analysis for the static design situation of the column is substantially lower than the design plastic moment needed by the column to satisfy the strength hierarchy. Since the column has been sized to bear the axial force for the seismic design situation together with the bending moment that satisfies the strength hierarchy, the column shall resist non-seismic load as well.

The equivalent section with the steel transformed into concrete is large. Consequently, stresses induced by service loads are always much less than the allowable stress at the serviceability limit states.

Consequently, other checks are not required; the column that results from using this method satisfies all the limit states. Therefore, this method provides an all-encompassing design solution.

The first example of the application section also shows how to use the proposed method in order to make a desired solution be optimal, which consists of an inverse use of the method. To that end, the loads and/or the spans have to be taken as variables. If the optimal cross-section from the direct use of the method turns out to be greater than the desired cross-section, the method can be used to determine how much to decrease the total load and/or the spans. The only question that remains is whether those decreases are possible or convenient.

## 7. Conclusions

The objective of the wider research that this paper stems from is the capacity design of seismic resistant RC frames. The activity is directed at analyzing this structural type and carrying out research to help reduce the incidences of the strength hierarchy on the proportioning of frame members, and to extend the operating horizons of RC frames in seismic design.

The paper focuses on the strong column–weak beam rule (also referred to as “minimum flexural strength of columns” or “column–beam moment ratio”) and addresses the issue of satisfying this rule with the minimum impact on the architectural design. To that end, the paper presents a simple method that provides the minimum cross-sectional dimensions that satisfy the strong column–weak beam rule with a reasonable amount of steel reinforcement. 

The method assumes that the beams have been optimized. If the structural designer has taken care to optimize each beam so as to minimize excess capacity, this method achieves the best result, because the design of the column (as of other frame elements) depends on the height of the beams framing into the joint and the amount of beam flexural reinforcement. If conversely the beams have been optimized according to an architectural criterion (e.g., the relationship with the floor), the result depends on the excess capacity of the beams.

Cross-sectional dimensions less than those furnished by this method require an amount of reinforcement that not only is very large but also makes it difficult to pour the concrete and to achieve good consolidation. Actually, the optimal cross-section allows a small reduction to be performed, but only if special column cages and concrete are used (which drastically affects the cost), while a greater reduction in any case would lead to poor quality constructional work.

Cross-sectional dimensions greater than those furnished by this method would allow a decrease in the steel reinforcement that, however, would be relatively small; as such, it would not be proportional to the increase in size.

Accordingly, the solution provided by this method is called the “optimal solution”, where optimal is meant in the sense that different cross-sections and reinforcements would be less satisfactory in term of construction quality, architectural design, or cost.

The method allows two unknowns to be determined. The direct way to use the method is to assume that the unknowns are the aspect ratio (or one side of the cross-section) and the amount of longitudinal reinforcement, while the other geometric and mechanical parameters are defined in advance (at the first stage of the structural design). The method can also be used in an indirect way by calibrating the design in order to make a desired RC column be optimal. Calibration involves adjusting the axial load in the column and the overstrength plastic moments of the beams, which in turn involves adjusting the dead and live loads of the floor, and the spans between the columns.

The completion of the design of the column requires dimensioning the hoops, which have to guarantee the hierarchy of strength of the failure modes. That is, the hoops have to suppress shear failures even under the most severe seismic shaking, so that failure is dictated by a suitable plastic mechanism (in this case, the hoop could be called “stirrup” since this reinforcement is used not only to confine the concrete, but also to resist shear in the column and to maintain lateral support for the reinforcing bars in regions where yielding is expected).

The optimal column according to the criterion established in this paper is a comprehensive solution, since it satisfies all the limit states, including the static ones. Thus, no verification is required.

## Figures and Tables

**Figure 1 materials-13-01919-f001:**
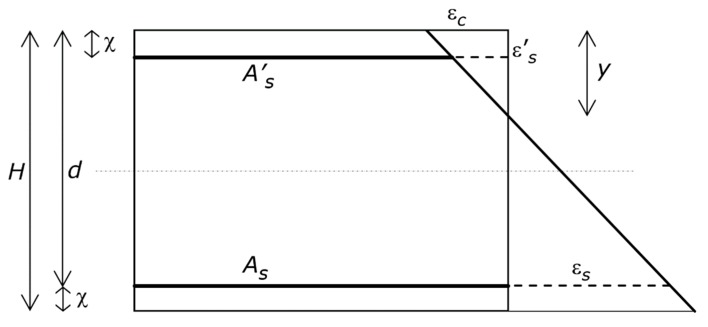
Reference structure. Geometric characteristics and strain profile, including the relevant symbols. The figure shows the depth of the neutral axis *y*, the tension and compression longitudinal reinforcements, whose areas are *A_s_* and *A^’^_s_*, respectively, the maximum strain in the concrete, which is ε*_c_*, and the strain in the tension and compression reinforcements, which are ε*_s_* and ε ^’^*_s_*, respectively.

**Figure 2 materials-13-01919-f002:**
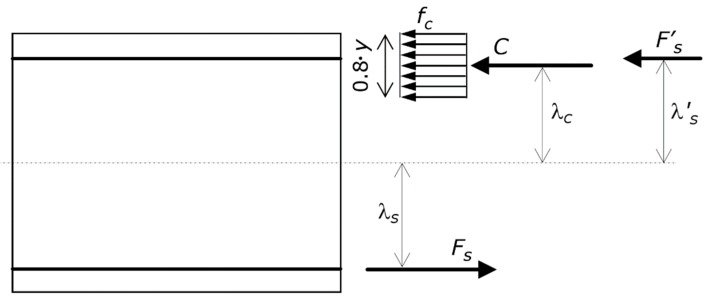
Reference structure. Ultimate stress profile on the cross-section. The figure shows the equivalent rectangular concrete compressive stress block *f_c_*, as well as its resultant internal force *C* together with its lever arm λ*_C_*, the resultant internal force *F ^’^_s_* in the compression longitudinal steel reinforcement together with its lever arm λ^’^*_s_*, and the resultant internal force *F_s_* in the tension longitudinal steel reinforcement together with its lever arm λ*_s_*.

**Figure 3 materials-13-01919-f003:**
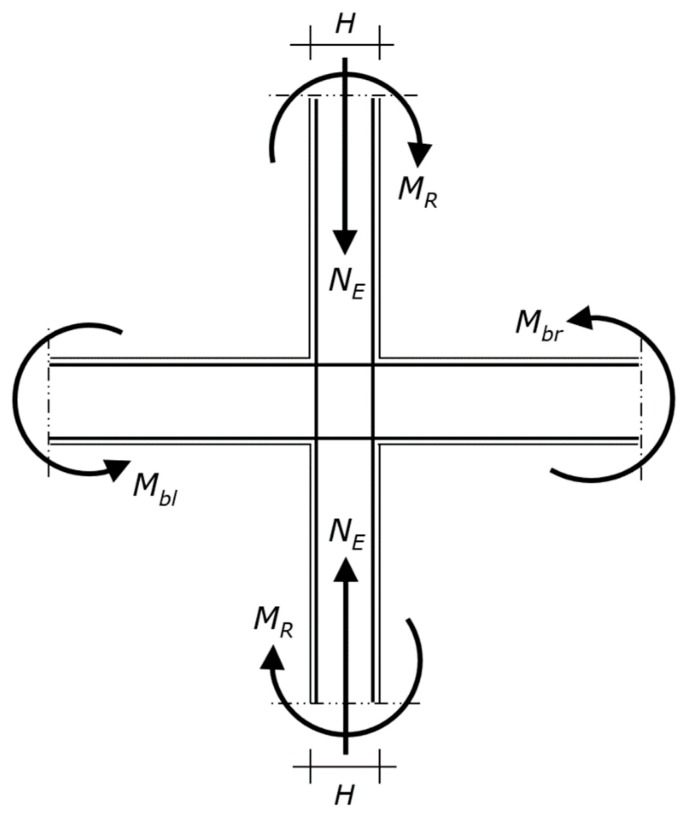
Diagram of the beam–column joint, where *M_bl_* and *M_br_* are the overstrength plastic moments that induce tension on the upper and lower fibers of the beam section, respectively (i.e., *M_bl_* and *M_br_* are obtained from the relevant design plastic moment of the beam multiplied by the overstrength factor). The bending moments *M_R_* are given by the rotational equilibrium. Since the upper column is equal to the lower column, *M_R_* = (*M_bl_* + *M_br_*)/2. The axial forces *N_E_* are the compressions that have to be transferred by the two columns together with *M_R_*.

**Figure 4 materials-13-01919-f004:**
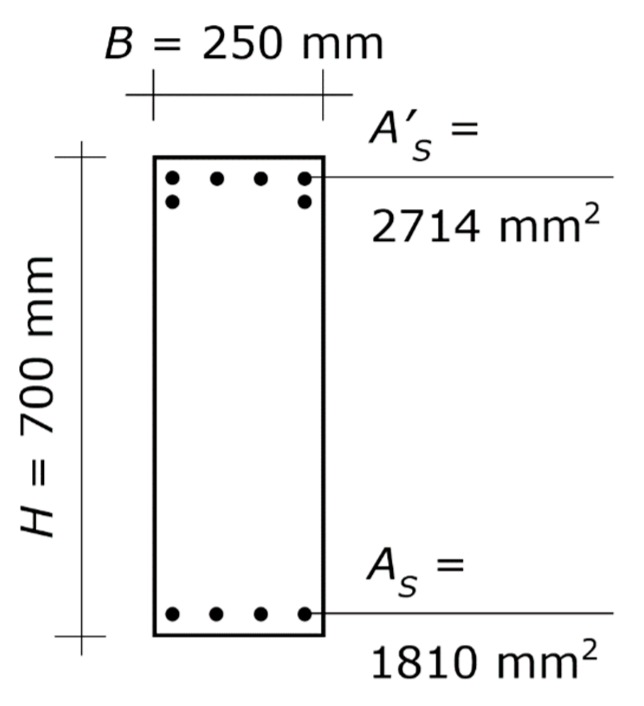
First application example. Cross-section of the two beams at the beam–column joint. The diagram shows the concrete section and the longitudinal steel reinforcement, while it does not show the stirrups (which play no direct role in column proportioning). The symbols are those defined in Section 2.1, used for the column. More specifically, *B* and *H* denote width and height of the beam, respectively, *A_s_* denotes the area of the bars at the lower edge, and *A^’^_s_* denotes the area of the bars at the upper edge.

**Figure 5 materials-13-01919-f005:**
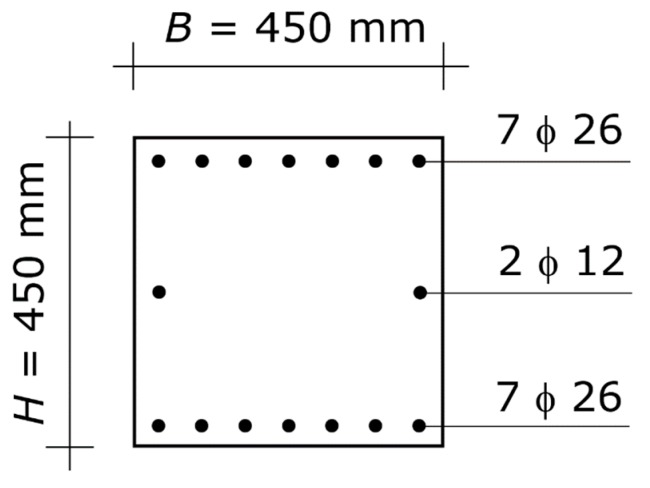
Optimal column of the first application example. Cross-section of the column. The diagram shows the concrete section and the longitudinal steel reinforcement, while it does not show the hoops (which shall be designed at a following stage of the structural design). The symbol ϕ denotes the diameter of the bars (it is the symbol used in Italy).

**Figure 6 materials-13-01919-f006:**
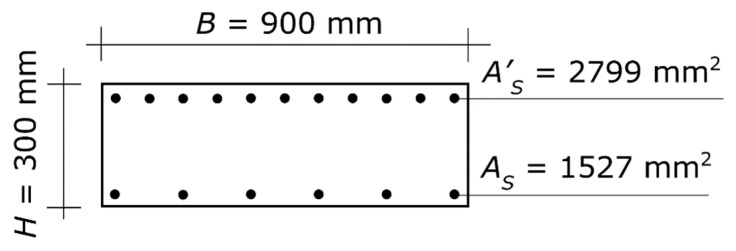
Second application example. Cross-section of the two beams at the beam–column joint. The diagram shows the concrete section and the longitudinal steel reinforcement, while it does not show the stirrups (which play no direct role in column proportioning). The symbols are those defined in Section 2.1.

**Figure 7 materials-13-01919-f007:**
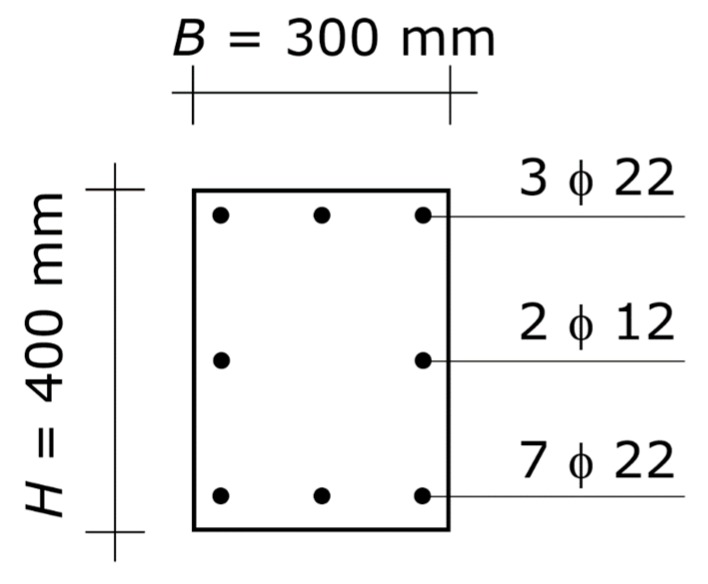
Optimal column of the second application example. Cross-section of the column. The diagram shows the concrete section and the longitudinal steel reinforcement, while it does not show the hoops (which shall be designed at a following stage of the structural design). The symbol ϕ denotes the diameter of the bars.

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
