# Peer review of "Optimal Design of Seismic Resistant RC Columns"

_materials, 2020, doi:10.3390/ma13081919_

Round 1
Reviewer 1 Report
Please see attachment.

Author Response
- The paper needs strengthening with the current state of the art and maybe some basic calculations on the economic consequences of the proposed method compared to the conventional one.
The revised version resubmitted has included evaluations about the cost, which the submitted manuscript had not considered.
- Abstract:
The abstract must be re-written in order to be attractive. Explaining the economy achieved by reducing reinforcement and section size in a column of a given size could give additional value to this research.
The abstract of the revised version resubmitted has been totally modified and above all has been enlarged. The new abstract addresses all the suggestions of the reviewers.
The impact on the cost has been considered throughout the whole article.
- Introduction:
It should also be explained why we are interested in smaller sections and of course at what spans; what is the maximum span achievable, because if we are to minimize section dimensions but need more columns both architectural design and construction costs will be challenged, again.
The Introduction of the revised version resubmitted is structured in a different way than that of the original submission.
In order to address that suggestion, the Introduction has been enlarged and subdivided into subsection.
Moreover, the new Introduction explains that the method can either consider that the spans are given or find the optimal spans.
- Trying to make the design more economic and possibly equally robust, a number of different methods have been proposed such as the compressive force path, which allows for less decongestion of reinforcement [1].
The revised version resubmitted has considered the paper that has been suggested, which has been cited and the new Introduction refers to it.
- Please try to enrich the introduction by giving more data on the current state of the art and omissions that the current research can cover.
The Introduction of revised version resubmitted has improved the state-of-the-art and the gaps that has to be filled.
- Line 229: the cover depends on the exposure conditions. What are the limits of the cover with this method?
The revised version resubmitted specifies that the concrete cover can be chosen by the structural designer, with no limits.
- Line 278: a full stop is missing after (Fig. 3)
Corrected.
- Line 442: a section on the economic consequences of the proposed method compared to the conventional one would be of interest.
The revised version resubmitted has included the cost among the aspects that are considered and the evaluations (in particular the connection between congestion of rebars and constructional cost)
Reviewer 2 Report
The paper deals with the strength hierarchy of structural elements of seismic resistant and proposes a method to design reinforced concrete column which satisfies the strong column-weak beam with minimum size. The paper is not well framed and requires some efforts for improvement.
What is the novelty of this method as compared to other in place practices?
How the results are comparable with other methods for designing RC with different assumptions?
The abstract should be one paragraph.
The abstract must be substantially improved.
Please be consistent in using Fig. or Figure
Why was a ɛcu of 0.35% used? The author must elaborate on that.
Author Response
- The paper is not well framed and requires some efforts for improvement.
The revised version resubmitted has been totally rewritten and one of the objective of the new version was to frame the presentation with a better logical flow.
- What is the novelty of this method as compared to other in place practices?
The revised version resubmitted has specified what the article adds to the subject and the novelty of the proposed method (in the new Introduction).
- How the results are comparable with other methods for designing RC with different assumptions?
The revised version resubmitted has provided comparisons between the results of this method and those obtainable with different assumptions (in particular, the role of concrete crushing strain).
- The abstract should be one paragraph.
Corrected. The new abstract is one paragraph only.
- The abstract must be substantially improved.
The abstract of the revised version resubmitted is totally different than that of the original submission. It has been enlarged and provides: the purpose of the work, the scope of the effort, the procedures used to execute the work, and the major findings. The abstract
- Please be consistent in using Fig. or Figure
Corrected. The revised version resubmitted uses "Fig." to refer to a specific figure (i.e., followed by a number) and "figure" when the term is used in general.
- Why was a ɛcu of 0.35% used? The author must elaborate on that.
The revised version resubmitted has provided literature to justify that assumption.
Moreover, the new version has overcome that assumption (it has removed that assumption by providing the possible range that the ultimate strain can range within, so that the methoud can be used for different values).
Reviewer 3 Report
The topic of the paper is interesting and suits the Journal of MDPI. However, minor revision is required before this manuscript is qualified to be published in this prestigious journal. The manuscript is needed to be revised grammatically. The authors are required to check the whole manuscript with a grammar specialist as it has several grammatical errors. Only after revising the manuscript based on the comments, the paper is suggested to be published in MDPI. Further information on various issues identified in the manuscript appears below:
- The authors have done a great job on the literature review. Please add more literature with regards to the works that have been published in the Journal of MDPI.
- Conclusion needs to be more concise. Please use less sentences containing percentages and illustrate the main conclusions in the manuscript. Please paraphrase your results and discussions and use them in the conclusion part.
Author Response
The author are required to check the whole manuscript with a grammar specialist as it has several grammatical errors.
The revised version resubmitted has been revised by a native English speaker.
The grammatical errors, some of which were typos, have been corrected.
- Please add more literature with regards to the works that have been published in the Journal of MDPI.
The revised version resubmitted has drastically improved the references and the cited papers include the main contribution in MDPI Journals about this topic (which are in Yellow).
- Conclusion needs to be more concise.
In order to address that suggestion, the conclusions of the revised version resubmitted focuses on the significant implications of the information presented in the body of the manuscript.
- Please use less sentences containing percentages and illustrate the main conclusions in the manuscript.
The revised version resubmitted has reduced the number of percentages. Moreover, the new conclusions follow logically from data presented.
- Please paraphrase your results and discussions and use them in the conclusion part.
The conclusions of the revised version resubmitted have considered what was in the Discussion.